# Antimicrobial and Cytotoxic Activities of Constituents from the Fruit of *Albizia lebbeck* L. Benth (Fabaceae)

**DOI:** 10.3390/molecules27154823

**Published:** 2022-07-28

**Authors:** Bosco Peron Leutcha, Jean Paul Dzoyem, Jean-Bosco Jouda, Denis Kehdinga Sema, Virginie Flaure Tsague Tankeu, Gabin Thierry Mbahbou Bitchagno, Billy Toussie Tchegnitegni, Flaure Rosette Ehawa Essoung, Bruno Ndjakou Lenta, Siméon Fogue Kouam, Florence Delie, Alain Meli Lannang, Norbert Sewald

**Affiliations:** 1Department of Chemistry, Faculty of Science, University of Maroua, Maroua P.O. Box 814, Cameroon; peron.leutcha@gmail.com (B.P.L.); devansema@gmail.com (D.K.S.); vtsague10@gmail.com (V.F.T.T.); 2Department of Chemistry, Higher Teacher Training College, University of Yaoundé I, Yaoundé P.O. Box 47, Cameroon; billytoussie@yahoo.fr (B.T.T.); lentabruno@yahoo.fr (B.N.L.); kfogue@yahoo.com (S.F.K.); 3Department of Biochemistry, Faculty of Science, University of Dschang, Dschang P.O. Box 67, Cameroon; jpdzoyem@yahoo.fr; 4Department of Chemical Engineering, School of Chemical Engineering and Mineral Industries, University of Ngaoundere, Ngaoundéré P.O. Box 454, Cameroon; joudabosco@yahoo.fr; 5Department of Chemistry, Faculty of Science, University of Dschang, Dschang P.O. Box 67, Cameroon; gabin1256@gmail.com; 6Department of Chemistry, Organic and Bioorganic Chemistry, Bielefeld University, 33501 Bielefeld, Germany; 7University Institute of Technology, University of Ngaoundere, Ngaoundere P.O. Box 455, Cameroon; rosflaure@yahoo.fr; 8Institute of Pharmaceutical Sciences of Western Switzerland, University of Geneva, Rue Michel-Servet 1, 1211 Geneva, Switzerland; florence.delie@unige.ch; 9Department of Chemistry, Higher Teacher Training College, University of Maroua, Maroua P.O. Box 55, Cameroon

**Keywords:** *Albizia lebbeck*, *Acacia*, Fabaceae, carbamide, flavonoids, antimicrobial activity, cytotoxicity

## Abstract

Twenty-two compounds were isolated from the fruit of *Albizia lebbeck* including one unprecedented, rare amino acid-derived zwitterionic and one new flavone derivative. The isolation was performed on repeated column chromatography over silica gel and their structures were determined by 1D-, 2D-NMR and HR-ESI-MS spectra together with reported data in the literature. The chemophenetic significance is also discussed. Some isolated compounds were reported for the first time to be found in the species. Additionally, compound **2** showed antibacterial activity and compounds **1** and **2** revealed moderate cytotoxic activity against the Raw 264.7 cancer cell line with IC_50_ values of 37.19 µM and 29.36 µM, respectively. Furthermore, a proposed biosynthetic pathway of compound **1** is described.

## 1. Introduction

*Albizia lebbeck* (or *Acacia lebbeck*) is a tree found in the Fabaceae family, mostly reported in tropical areas and in rainfall forests and usually grows in lateritic and sandy soils [1]. During our ethnobotanical survey in Northern Cameroon, *Albizia lebbeck* was claimed by the local population to treat dysentery, asthma, hemorrhoids, bronchitis, eczema, leprosy, human fertility, and diarrhea, meanwhile some constituents of *A. lebbeck* are reported for their antimicrobial, antioxidant, anti-inflammatory, and toxicity properties [2,3,4,5,6,7]. In light of our study and as part of our ongoing research on bioactive metabolites from Cameroonian medicinal plants [8,9], we isolated a new amino acid-derived zwitterionic (**1**) and a new flavone flavonoid derivative (**2**) together with twenty known compounds from the fruit of *A. lebbeck*. Eugenol (**5**) is reported for the first time from the Fabaceae family while 29-hydroxyhopane (**8**), spinassterol (**16**), spinasterol-3-O-*β*-*D*-glucopyranoside (**17**), *D*-mannitol (**20**), octacosanoic acid (**21**), and bis ((*S*)-2, 3-dihydroxypropyl) (**22**) are reported for the first time from *A. lebbeck*. Herein, we report the isolation, structure elucidation, antimicrobial potential, as well as cytotoxic properties of the extracts and isolated compounds.

## 2. Results

Column chromatography (CC) of the CH_2_Cl_2_/MeOH (1/1, *v*/*v*) extract of the seeds and pods of *A. lebbeck* led to the isolation and structure elucidation of one unprecedented amino acid-derived zwitterionic named chiakine (**1**), one new flavone derivative, lebbeckisoetin A (**2**), and twenty known compounds (**3**–**22**) (Figure 1).

### 2.1. Structure Elucidation of Compounds **1** and **2**

Compound **1** was obtained as an optically white powder which exhibits a levorotatory activity with ${\left[\alpha \right]}_{D}^{20}$ = −1.4 (c 0.5, MeOH). Its HR-ESI-MS showed a pseudomolecular ion peak [M+Na]^+^ at *m/z* 170.0542 (Appendix A) (calcd. *m/z* 170.0536 for C_4_H_9_N_3_O_3_Na) indicating two double-bond equivalents. The IR spectrum displayed absorption bands of OH (3281 cm^−1^), NH (2929 cm^−1^), ester/amide carbonyl (1634 cm^−1^) and C-O/C-N (1326/1157 cm^−1^). Its ^13^C-NMR data (Table 1) together with the HSQC spectrum (Appendix A) led to the identification of four carbon atoms including two carbonyls at *δ*_C_ 172.4 and 161.6, one *sp*^3^ methine at *δ*_C_ 55.4, and one *sp*^3^ methylene at *δ*_C_ 40.2. The above data (Table 1) suggested that compound **1** had aminomethine and aminomethylene groups [10]. Its ^1^H-NMR spectrum (Appendix A) exhibits signals of three groups of protons at *δ*_H_ 3.55 (ddd, J = 15.3, 3.6, 1.3 Hz, H-1′a), 3.35 (ddd, J = 15.2, 6.5, 1.4 Hz, H-1′b) and 3.70 (ddd, J = 6.5, 3.5, 1.3 Hz, H-2′). The COSY spectrum (Appendix A), led to the location of the *vicinal* as well as *geminal* correlations on the structure: H-1′a (*δ*_H_ 3.55) and H-1′b (*δ*_H_ 3.35) (*gem*), and H-2′ (*δ*_H_ 3.70), H-1′a (*δ*_H_ 3.55) and H-1′b (*δ*_H_ 3.35) (*vic*) (Figure 2). The HMBC spectrum (Appendix A) led to position both quaternary carbons through the correlation between H-1′a/H1′b (*δ*_H_ 3.55/3.35) with C-1 (*δ*_C_ 161.6) and C-3′ (*δ*_C_ 172.4) (Figure 2). Thus, compound **1** was identified as a new amino acid-derived zwitterionic natural product trivially named chiakine. Compound **1** would be derived from the hypothetical precursors carbamic acid (**23**) and phosphoseriamide (**27**). In this proposed pathway, the phosphoseriamide could be formed from the nucleotide precursor glycoaldehyde (**24**) [11] and glycolaldehyde-2-phosphate (**25**) via the intermediate glycolaldehyde phosphate aminonitrile (**26**) [12].

Compound **2** was obtained as an optically yellow amorphous powder which exhibits a dextrorotatory activity with ${\left[\alpha \right]}_{D}^{20}$ = +164.7 (c 0.5, MeOH). Its molecular formula C_23_H_24_O_12_ was established by the positive ion mode HR-ESI-MS which showed the pseudomolecular ion peak [M+H]^+^ at *m/z* 493.1336 (Appendix A) (calcd. *m/z* 493.1346 for C_23_H_25_O_12_) containing 12 double-bond equivalents. The IR spectrum displayed absorption bands of the hydroxyl group (3852 cm^−1^), carbonyl group (1733 cm^−1^), and C-O group (1206 cm^−1^). Its ^1^H-NMR spectrum (Appendix A) reveals the signals of a chelated proton at *δ*_H_ 13.05 (s) and a phenolic proton at *δ*_H_ 9.09 (s) both attributed to polyphenols. The signals of a singlet at *δ*_H_ 7.17 (1H) together with two doublets of a metasystem of a proton at *δ*_H_ 6.38 (d, *J* = 2.2 Hz) and 6.71 (d, *J* = 2.2 Hz) are characteristic of a tetrasubstituted ring A of a flavone derivative [13]. Moreover, the metasystem of the previously described protons is supported by the COSY spectrum (Appendix A) with correlation peaks between H-6 (*δ*_H_ 6.38) and H-8 (*δ*_H_ 6.71). In addition, two singlet signals are observed at *δ*_H_ 6.99 (1H, s, H-3′) and *δ*_H_ 7.37 (1H, s, H-6′) suggesting a pair of paraprotons in ring B, two singlets of methoxy protons at *δ*_H_ 3.87 (3H, s, MeO-7) and *δ*_H_ 3.86 (3H, s, MeO-4′). The ^1^H-NMR signal at *δ*_H_ 5.05 (d, *J* = 7.2 Hz) was assigned to the *β*-anomeric proton characteristic of a sugar moiety [14]. The ^13^C-NMR (Appendix A) together with DEPT-135 spectra reveal the signals of 23 carbon atoms, where the signals of a sugar moiety were identified as glucopyranosyl at *δ*_C_ 101.6 (C-1″), 73.9 (C-2″), 77.9 (C-3″), 70.5 (C-4″), 77.5 (C-5″), and 61.3 (C-6″) [8] together with those of two methoxy groups at *δ*_C_ 56.6 (MeO-7) and 56.5 (MeO-4′). The aglycone was identified from the aforementioned observation as a flavone flavonoid derivative with characteristic signals at *δ*_C_ 161.5 (C-2), 109.6 (C-3), 182.6 (C-4), 161.6 (C-5) 165.6 (C-7), 157.8 (C-8a), 150.6 (C-2′), and 151.9 (C-4′) [15,16]. Extensive analysis of COSY, HSQC, and HMBC (Figure 3) spectra allowed the connection of the functional groups to be assigned and the structure of compound **2** to be built. Furthermore, the cross correlation observed in the HMBC spectrum (Appendix A) between the methoxy protons at *δ*_H_ 3.87 (MeO-7) and *δ*_H_ 3.86 (MeO-4′) with carbon at *δ*_C_ 165.6 (C-7) and 151.9 (C-4′), respectively showed that there were linked at (C-7) and (C-4′). The linkage of the sugar moiety in the aglycone was assumed by the correlation between the anomeric proton at *δ*_H_ 5.05 (H-1″) and carbon at *δ*_C_ 150.6 (C-2′) and this finding was further confirmed by the NOESY spectrum. Indeed, extensive analysis of the NOESY spectrum showed a strong correlation between H-1″ (*δ*_H_ 5.05) with H-3 (*δ*_H_ 7.17) and H-3′ (*δ*_H_ 6.99), then between H-3′ (*δ*_H_ 6.99) and H-6″ (*δ*_H_ 3.74/3.43) (Figure 3), which unambiguously supports the position of sugar moiety at C-2′. Therefore, the structure of compound **2** was characterized as a new flavone derivative trivially named Lebbeckisoetin A.

The known compounds were identified based on the interpretation of their NMR (1D and 2D) spectra and confirmed by comparing the data to those described so far in the literature as: quericitrin (**3**) [9], quercetin-3-*O-β-D*-glucopyranoside (**4**) [16], eugenol (**5**) [3], *(E)-p*-coumaric acid (**6**) [17], *(Z)-p*-coumaric acid (**7**) [18], 29-hydroxyhopane (**8**) [19], lupeol (**9**) [20], betulin (**10**) [21], oleanolic acid (**11**) [22], lactulose or *α-D*-galactopyranoside-(1→2′)-*β-D*-fructofuranoside (**12**) [23], saccharose or *α-D*-glucopyranosyl-(1→2′)-*β*-D-fructofuranoside (**13**) [24], stigmasterol (**14**) [25], stigmasitosterol-3-*O-β-D*-glucopyranoside (**15**) [26], spinassterol (**16**) (Kamboj and Saluja, 2011), spinasterol-3-O-*β*-*D*-glucopyranoside (**17**) [27], *β*-sitosterol (**18**), *β*-sitosterol-3-*O-β-D*-glucopyranoside (**19**) [4,27], *D*-mannitol (**20**) [28], octacosanoic acid (**21**) [29], and hexacosan-1′,26′-dioate of bis ((*S*)-2, 3-dihydroxypropyl) (**22**) [30].

### 2.2. Physicochemical Characteristics of Compounds **1** and **2**

Chiakine (**1**) white powder, ^1^H-NMR, and ^13^C-NMR (D_2_O, 600 MHz), see Table 1; HR-ESI-MS at *m/z* 170.0542 [M + Na]^+^ and *m/z* 148.0728 [M + H]^+^ (calcd. *m/z* 170.0536 for C_4_H_9_N_3_O_3_Na^+^ and *m/z* 148.0722 for C_4_H_10_N_3_O_3_^+^).

Lebbeckisoetin A (**2**) yellow amorphous powder, ^1^H-NMR, and ^13^C-NMR (DMSO-*d_6_*, 600 MHz), see Table 2; HR-ESI-MS at [M + H]^+^ at *m/z* 493.1336 (calcd. *m/z* 493.1346 for C_23_H_25_O_12_) and [M + Na]^+^ at *m/z* 515.1156 (calcd. *m/z* 515.1165 for C_23_H_24_O_12_Na^+^).

### 2.3. Biological Assays

The antimicrobial activities were performed on the pod crude extract and several isolated compounds using ketoconazole and ciprofloxacin as antifungal and antibacterial drug references. The interpretation of the result was based on the positive controls [31]. The CH_2_Cl_2_-MeOH (1:1, *v*/*v*) crude extract, *n*-hexane, ethyl acetate, and *n*-butanol extracts together with compounds **1**, **2**, **12,** and **13** were tested against five microbial strains (Table 2). All the tested samples showed antimicrobial activity against at least one microbial strain, except the *n*-butanol extract which did not show any antimicrobial activity. Compounds **1** and **2** showed the most potent antifungal activity with MIC values of 32 µg/mL against *Candida albicans* while compound **2** further showed similar antibacterial activity against *Escherichia coli*. Compounds **12** and **13** were the least active in both bacteria and yeast strains. It clearly appears that the antimicrobial activity of *Albizia lebbeck* pods against *Candida albicans* and *Escherichia coli* increased with fractionation, from the CH_2_Cl_2_-MeOH (1:1, *v*/*v*) crude extract (MIC values of 512 µg/mL and 256 µg/mL) to the ethyl acetate extract (MIC values of 128 µg/mL) and finally to compounds **1** and **2,** which were the most active (MIC values of 32 µg/mL). The above results are in accordance with previous reports of antimicrobial assays on the seeds, pods, and flowers of *A. lebbeck* [10,32] and could support the local uses of the fruit of this plant for the treatment of child and adult diarrhea. The mouse macrophages raw 264.7 cancer cell line was used to investigate the in vitro cytotoxicity of compounds **1** and **2** using the WST-1 assay. The concentration–response graph (Figure 4) showed that compounds **1** and **2** inhibited the proliferation of Raw 264.7 cancer cells in a dose-dependent manner. The IC_50_ values were determined based on the percentage of viable cells less than 50%. The two compounds showed moderate cytotoxic activity against Raw 264.7 with IC_50_ values of 37.19 µM and 29.36 µM, respectively. Raw 264.7 cell line has been exhaustively used for the screening of secondary metabolites for their cytotoxic activities [33,34]. However, the high sensitivity of Raw 264.7 compared to other cancer cell lines, such as human leukemia monocyte cell line THP-1 and the human lung adenocarcinoma A549 cell lines, has been previously reported [8]. To the best of the author’s knowledge, the anticancer properties of an isolated compound from the seeds of *A. lebbeck* have never been reported. The obtained activity of chiakine (**1**) and lebbeckisoetin A (**2**) on Raw 264.7 cells indicated that these compounds might be potential candidates for new anticancer agents.

## 3. Discussion

Twenty-two (**1**–**22**) compounds were isolated using chromatographic techniques from the crude extracts of the fruit of *Albizia lebbeck* L. Benth, including a new carbamide (**1**) and flavone (**2**) derivatives together with twenty known compounds grouped into two flavonoids (**3**, **4**), three alkylphenols (**5**–**7**), four triterpenoids (**8**–**11**), three sugars (**12**, **13**, **20**), three sterols (**14**, **16**, **18**), three saponins (**15**, **17**, **18**), and two fatty acids (**21**, **22**). To the best of our knowledge, compounds (**1**, **2**, **5**, **8**, **16**, **17**, **20**, **21**, and **22**) are herein reported for the first time from *A. lebbeck*. This is the first report of compound **5** in either the Mimosaceae subfamily or the Fabaceae family. Compound **5** is considered a chemomarker of *Syzygium aromaticum*, well known as “*Clou de Girolf*” [3]. However, compounds **1** and **2** were found to be new and are reported for the first time from the Fabaceae family as their derivatives were reported from *A. lebbeck* [10,35]. This is the first report of compounds **8**, **16**, **17**, **20**, **21**, and **22** from *Albizia lebbeck*, although they have already been reported from the *Albizia* genus as **8** (*Albizia chinensis*) [36], **16**–**17** (*Albizia dealbata* and *Albizia melanoxylon*) [28] and **20**–**22** (*Albizia Amara* and *Albizia julibrissin*) [37,38]. This paper has then strengthened the chemophenetic survey on the *Albizia* genus. These observations give new insights into the occurrence of flavonoids and saponins in the genus *Albizia* and the family Fabaceae. The flavone flavonoids might represent a new significant chemophenetic finding in the *Albizia* genus, specifically in the *Albizia lebbeck* plant. However, saponins in general, could be considered chemotaxonomic markers for the genus *Albizia* due to the soapy aspect of all the species. The study of the fruit of *A. lebbeck* L. Benth led to the isolation of a chemical constituent which enriches the information on the phytochemistry of the plant and provides further knowledge in regard to the possible chemotaxonomic markers present in the *A. lebbeck* species, the *Albizia* genus, as well as the Fabaceae family. It is notable that compounds **1**, **2**, and **5** have not yet been reported either from the *Albizia* genus or the Fabaceae family and suggest a new observation of chemotaxonomic knowledge of the *Albizia* and more diverse chemical constituents from *A. lebbeck* L. Benth.

## 4. Materials and Methods

### 4.1. General Experimental Procedures

The melting points were measured on a Gallenkamp melting point apparatus. Optical rotations were recorded with a JASCO DIP-360 polarimeter. The IR spectrum was measured in MeOH on a JASCO A-302 spectrophotometer. The proton and carbon (1D and 2D)-NMR (600/500 MHz) spectra were measured on a Bruker AMX machine. The chemical shifts of proton and carbon were recorded based on the internal reference TMS (Tetramethylsilane) in *δ* (ppm). Moreover, coupling constants (*J*) were measured in Hz. The ESIMS was recorded on a double-focusing mass spectrometer (Varian MAT 311A), while HREIMS was recorded on a JEOL HX 110 mass spectrometer. Silica gel 60 (0.2–0.5 mm and 0.2–0.063 mm) (70–230 and 240–300 mesh sizes, E. Merck) was used as a stationary phase for CC (Column Chromatography). The purity of compounds and the monitoring of fractions were based on precoated silica gel TLC (Thin Layer Chromatography) plates supported on either plastic or aluminum sheets (E. Merck, F_254_). Spots were visualized on TLC with UV light (254 nm and 365 nm) on a CN-6 UV spectrometer (made in France) then sprayed with ceric sulphate and heated at about 90 °C.

### 4.2. Plant Material

The fruit of *A. lebbeck* was harvested on 20 April 2018 in Maroua town, headquarters of the Far-North Region of Cameroon and identified by Mr Tadjouteu Flubert by comparing it with a specimen available at the National Herbarium at Yaoundé with a Voucher specimen 58964/NHC.

### 4.3. Extraction and Isolation

The plant material used for this work was the mature and dry fruit of *Albizia lebbeck.* The pale-yellow seeds of *A. lebbeck* were powdered (1.3 kg) and extracted for 72 h with CH_2_Cl_2_/MeOH (1/1, *v*/*v*) at room temperature. The mixture was filtered and taken to dryness under reduced pressure at about 60 °C to afford a crude extract of 120.3 g. Compound **1** (2.1 g) was precipitated in the crude extract and was obtained by filtration. Then, 100.0 g of this extract were subjected to open CC over silica gel and eluted with a gradient of *n*-hexane/CH_2_Cl_2_/MeOH to afford four major fractions: A (1.2 g), B (1.9 g), C (60.5 g), and D (10.8 g). Fraction A (1.2 g) was purified using CC (isocratic, 15 % *n*-hexane/CH_2_Cl_2_) to afford **21** (15 mg) and **8** (11 mg). Fraction B (1.9 g) was also subjected to open CC (gradient, CH_2_Cl_2_/EtOAc/MeOH) which led to **9** (19 mg), **5** (89 mg), **10** (10 mg), **11** (20 mg), and **20** (**17** mg). Fraction C (60.5 g) was submitted to open CC (gradient, EtOAc/MeOH) to yield **6** (150 mg), **7** (56 mg), **12** (705 mg), and **13** (969 mg). Fraction D (10.8 g) was purified using open CC (silica gel, gradient, EtOAc/MeOH) which led to **19** (704 mg), **15** (908 mg), and **17** (431 mg).

Furthermore, 10.3 kg of powdered pods of *A. lebbeck* were extracted for 72 h with CH_2_Cl_2_-MeOH (1/1, *v*/*v*). The mixture was filtered and taken to dryness under 300 mbar at 40 °C to afford 500.6 g of crude extract. Then 480.0 g of crude extract were further successively extracted with *n*-hexane, ethyl acetate, and *n*-butanol to afford 203.8 g, 23.5 g, and 109.9 g of residues, respectively. A portion of the *n*-hexane extract (200.9 g) was subjected to CC (gradient, *n*-hexane/CH_2_Cl_2_) to afford **18** (90.2 mg), a mixture of **14** and **16** (105.1 mg), and **22** (80.2 mg). Next 20.0 g of ethyl acetate extract was subjected to CC (*n*-hexane/CH_2_Cl_2_/EtOAc/MeOH) to yield four fractions F (2.0 g), G (2.9 g), H (6.5 g), and I (3.8 g). Finally, fraction I (3.8 g) was run over open CC (EtOAc/MeOH) to afford **2** (5 mg), **17** (5 mg), **3** (7 mg), and **4** (4.3 mg).

### 4.4. Antimicrobial Assay

Five microorganisms from American Type Culture Collection including four bacterial (two Gram-positive and two Gram-negative) and one fungal strain were used: *Candida albicans* ATCC 9028, *Staphylococcus aureus* (ATCC 1026), *Enterococcus faecalis* (ATCC 29212), *Escherichia coli* (ATCC 25922), and *Pseudomonas aeruginosa* (ATCC 74117). The antibacterial activity of the crude extract, fractions, and compounds was assessed by determining the minimum inhibitory concentration (MIC) and minimum bactericidal and fungicidal concentration (CBC and MFC) using the broth microdilution method, as previously described [39]. Briefly, the MIC was performed by broth microdilution method, with Mueller–Hinton broth for bacteria and Sabouraud dextrose broth for yeast. Stock solutions of samples were prepared in 100% dimethyl sulfoxide and twofold serial dilutions were prepared in media in amounts of 100 μL per well in a 96-well. Microbial suspensions were prepared in culture media, and 100 μL of this inoculum was added to each well of the plate, resulting in a final inoculum of 1.5 × 10^6^ CFU/mL for bacteria and 2 × 10^4^ CFU/mL for yeast. The final concentration of samples ranged from 2 μg/mL to 256 μg/mL and from 8 μg/mL to 1024 μg/mL for extracts. The medium without the agents was used as a growth control and the blank control used contained only the medium. Ciprofloxacin and ketoconazole served as the standard drug controls. The MIC of samples was detected after 24 h (for bacteria) and 48 h (for fungi) of incubation at 37 °C, following an addition (40 µL) of 0.2 mg/mL of p-iodonitrotetrazolium (INT) chloride and incubation at 37 °C for 30 min, as the lowest sample concentration that prevented the color change of the medium and exhibited complete inhibition of microbial growth. The MBC or MFC was determined by adding 50 µL aliquots of the preparations, which did not show any growth after incubation during MIC assays to 150 µL of adequate broth. These preparations were incubated at 37 °C for 48 h. The MBC or MFC was regarded as the lowest concentration of a sample which did not produce a color change after an addition of INT, as mentioned above.

### 4.5. Cytotoxicity Assay

The cytotoxicity of unprecedented carbamide (**1**) and flavone (**2**) derivatives were evaluated on the Raw 264.7 cells (ATCC) cancer cell line. Cells were cultured in DMEM (Dulbecco’s Modified Eagle’s Medium) culture media supplemented with 10% fetal calf serum and 1% antibiotics (100 IU/mL penicillin and 100 μL/mL streptomycin) and maintained at 37 °C in a humidified atmosphere containing 5% CO_2_. The WST-1 assay was used to quantify the cell viability and the cytotoxicity was evaluated by determining the concentration inhibiting 50% of viable cells (IC_50_), as previously described [8].

## Figures and Tables

**Figure 1 molecules-27-04823-f001:**
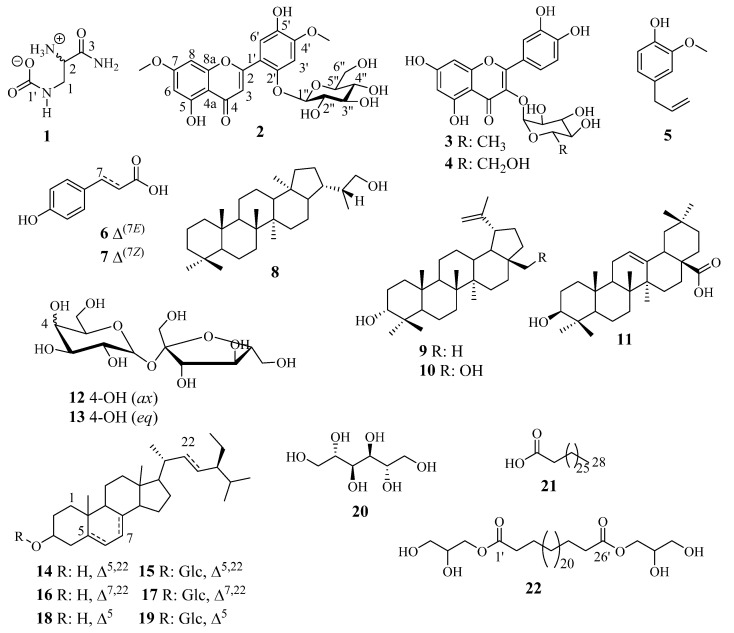
Structures of compounds **1**–**22**.

**Figure 2 molecules-27-04823-f002:**
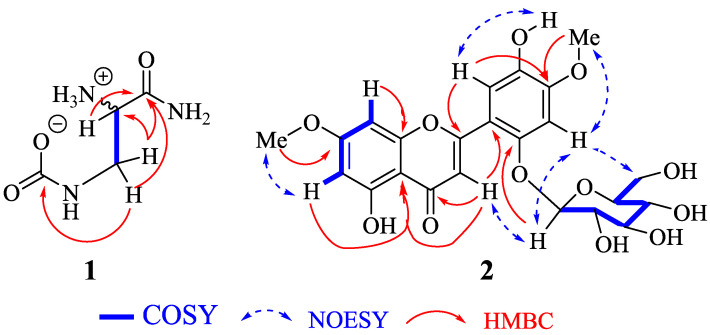
COSY, HMBC, and NOESY correlations of compounds **1** and **2**.

**Figure 3 molecules-27-04823-f003:**
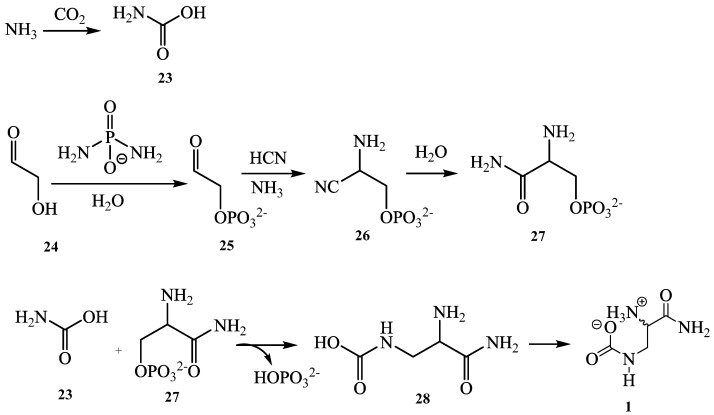
Proposed biosynthetic pathway of compound **1**.

**Figure 4 molecules-27-04823-f004:**
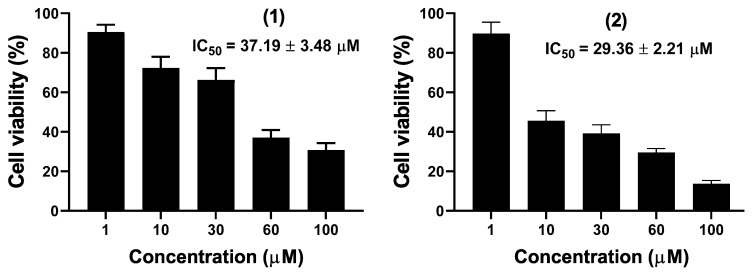
Cytotoxicity of compounds (**1**) and (**2**) against Raw 264.7 cancer cell line.

**Table 1 molecules-27-04823-t001:** ^13^C and ^1^H-NMR spectroscopic data (^1^H 600 MH_Z_, ^13^C 150 MH_Z_ for compounds **1** (in D_2_O) and **2** (in DMSO-*d*_6_).

N°	1			N°	2		
	*δ* _C,_	*δ*_H_ (m, *J* in Hz)	HMBC		*δ* _C,_	*δ*_H_ (m, *J* in Hz)	HMBC
1′	161.6			1			
1	40.2	3.55 (ddd, 15.3, 3.6, 1.3)	1, 2′, 3′	2	161.5		
		3.35 (ddd, 15.2, 6.5, 1.4)					
2	55.4	3.70 (ddd, 6.5, 3.5, 1.3)	1′, 3′	3	109.6	7.17 (s)	2, 4, 4a, 1′
3	172.4			4	182.6		
				4a	105.1		
				5	161.6		
				6	98.3	6.38 (d, 2.2)	4a, 5, 7, 8
				7	165.6		
				8	92.8	6.71 (d, 2.2)	4a, 6, 7, 8a
				8a	157.8		
				1′	111.6		
				2′	150.6		
				3′	101.1	6.99 (s)	1′, 2′, 4′, 5′
				4′	151.9		
				5′	141.5		
				6′	114.4	7.37 (s)	2, 2′, 4′, 5′
				7-OMe	56.6	3.87 (s)	7
				4′-OMe	56.5	3.86 (s)	4′
				1″	101.6	5.05 (d, 7.2)	2′
				2″	73.9	3.35 (m)	
				3″	77.9	3.43 (dd, 9.6, 4.0)	
				4″	70.5	3.14 (td, 8.6, 4.8)	
				5″	77.5	3.29 (m)	
				6″	61.3	3.74 (dd, 9.7, 5.4) 3.43 (dd, 9.6, 4.0)	
				4′-OH		9.09 (s)	4′
				5-OH		13.05 (s)	

d: doublet, dd: doublet of doublets, ddd: doublet of doublet of doublets, m: multiplet, s: singlet, td: triplet of doublets.

**Table 2 molecules-27-04823-t002:** Minimum Inhibitory Concentration (MIC), Minimum Bactericidal Concentration (MBC) and Minimal Fungicidal Concentration (MFC) values (in µg/mL) of extracts, fractions, and compounds from *Albizia lebbeck* fruit.

Samples	Microorganisms									
	*Candida albicans*		*Escherichia coli*		*Staphylococcus aureus*		*Pseudomonas aeruginosa*		*Enterococcus faecalis*	
	MIC	MFC	MIC	MBC	MIC	MBC	MIC	MBC	MIC	MBC
CH_2_Cl_2_-MeOH (1:1, *v*/*v*) extract	512	-	128	-	512	-	256	-	256	1024
*n*-hexane extract	-	-	-	-	512	-	-	-	512	-
ethyl acetate extract	128	-	128	1024	256	-	512	-	256	1024
*n*-butanol extract	-	-	-	-	-	-	-	-	-	-
**1**	32	256	256	-	128	-	128	-	-	-
**2**	32	256	32	256	256	256	128	256	64	-
**12**	256	-	-	-	256	-	-	-	256	-
**13**	-	-	-	-	256	-	-	-	-	-
*Ketoconazole*	0.5	64	-	-	-	-	-	-	-	-
*Ciprofloxacin*	-	-	2	512	1	512	2	8	4	4

- = >1024 µg/mL for crude extracts and fractions and >256 µg/mL for compounds. Four bacteria strains and one yeast strain from American Type Culture Collection were used: *Candida albicans* ATCC 9028, *Staphylococcus aureus* (ATCC 1026), *Enterococcus faecalis* (ATCC 29212), *Escherichia coli* (ATCC 25922), and *Pseudomonas aeruginosa* (ATCC74117).

## Data Availability

The raw data supporting the conclusions of this article will be made available by the authors without undue reservation to any qualified researcher.

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
