# Peer review of "Antimicrobial and Cytotoxic Activities of Constituents from the Fruit of Albizia lebbeck L. Benth (Fabaceae)"

_molecules, 2022, doi:10.3390/molecules27154823_

Round 1
Reviewer 1 Report
Twenty-two compounds were isolated from the fruits of Albizia lebbeck, including one unprecedented rare amino acid-derived zwitterionic and one new flavone derivative. Some isolated compounds were reported for the first time to be found in the species. Also, compound 2 shows antibacterial activity and compounds 1 and 2 revealed moderate cytotoxic activity in a cancer cell line.
This manuscript has certain novelty and uniqueness, but it still needs minor revision:
1. in the introduction, the author should cite references about the uses of A. lebbeck in local, and the references to pharmacological effects of known components of A. lebbeck.
2. As the description in the methods, the plant material used for this work was the dry fruits of Albizia lebbeck. To my best knowledge, the natural ingredients contained in the plant, especially fruits and seeds, are strongly linked to their maturity at harvest time. the author may want to describe this detail.
Author Response
Reviewer 1
Twenty-two compounds were isolated from the fruits of Albizia lebbeck, including one unprecedented rare amino acid-derived zwitterionic and one new flavone derivative. Some isolated compounds were reported for the first time to be found in the species. Also, compound 2 shows antibacterial activity and compounds 1 and 2 revealed moderate cytotoxic activity in a cancer cell line.
Thank you for your pertinent suggestions. This section has been revised accordingly.
This manuscript has certain novelty and uniqueness, but it still needs minor revision:
- in the introduction, the author should cite references about the uses of A. lebbeck in local, and the references to pharmacological effects of known components of A. lebbeck.
This requested information have been added in the manuscript.
- As the description in the methods, the plant material used for this work was the dry fruits of Albizia lebbeck. To my best knowledge, the natural ingredients contained in the plant, especially fruits and seeds, are strongly linked to their maturity at harvest time. the author may want to describe this detail.
Details regarding the maturity of fruits at harvest time have been provided in the manuscript.
Reviewer 2 Report
Antimicrobial and cytotoxic activities of constituents from the fruits of Albizia lebbeck L. Benth (Fabaceae)
Review comments
Abstract
Line 26- Is the number ‘2’ referring to Compound 2? If so, it should please be stated for clarity.
Introduction
Lines 36-38- Please provide a reference or references for the uses of Albizia lebbeck listed.
Table 1- Please indicate beneath the table the meaning of the letters in the Table. For example, m-multiplet d-doublet e.t.c.
Lines 263-269- Please indicate the full meaning and process of determination of MFC values obtained in Table 2. Very little was said about the MFC.
Manuscript has some minor editing to be done .

Author Response
Reviewer 2
Antimicrobial and cytotoxic activities of constituents from the fruits of Albizia lebbeck L. Benth (Fabaceae)
Abstract
Line 26- Is the number ‘2’ referring to Compound 2? If so, it should please be stated for clarity.
Thank you for the remark, the missing word has been added
Introduction
Lines 36-38- Please provide a reference or references for the uses of Albizia lebbeck listed
The references have been added in the manuscript.
Table 1- Please indicate beneath the table the meaning of the letters in the Table. For example, m-multiplet d-doublet e.t.c.
Done.
Lines 263-269- Please indicate the full meaning and process of determination of MFC values obtained in Table 2. Very little was said about the MFC.
Thank you for the concern, a detailed paragraph has been added in the manuscript regarding the MFC.
Manuscript has some minor editing to be done.
The manuscript has been cross checked accordingly
Line 151: The positive controls should be mentioned at the end of the sentence.
Author’s response: The requested statement has been added.
Line 164: Consider reframing the sentence to ...... are in accordance with previous reports of antimicrobial assays on the seeds.......
Author’s response: The requested correction has been made.
Line 166: Authors should confirm if Raw 264.7 for sure are CANCER cell lines. They are mouse macrophage cells for determining toxicity or selective toxicity of a compound/extract generally but are they CANCER cell lines? Please confirm.
Author’s response: Yes Raw 264.7 is effectively a cancer cell line (RAW 264.7 is a macrophage cell line that was established from a tumor in a male mouse induced with the Abelson murine leukemia virus).
Dose-dependent should be hyphenated througout the document
Author’s response: The requested correction has been made.
The abbreviation MIC has been defined but MFC not defined.
Author’s response: MFC has been defined , please see line 184.
Gram-positive ad Gram-negative should be hyphenated throughout the document.
Author’s response: The requested correction has been made.
Reviewer 3 Report
Line 40-46: This section sounds more like conclusion.
Line 47: Should be results only since discussion is in another section.
Line 150: 'some' change to several isolated compounds.
Line 181: Why was Raw 264.7 cancer cell line used instead of a human cell line since the intended target is to establish potential chemotherapeutic properties.
Table 2: Unit has to be provided in table 2 to accurate interpret the results.
Line 193: Two fatty what? Fatty acids?

Author Response
Reviewer 3
Line 40-46: This section sounds more like conclusion.
This common section generally summarizes the new finding of the reasearch work carried out.
Line 47: Should be results only since discussion is in another section.
Thanks. This has been corrected accordingly
Line 150: 'some' change to several isolated compounds.
Corrected as requested
Line 181: Why was Raw 264.7 cancer cell line used instead of a human cell line since the intended target is to establish potential chemotherapeutic properties.
We agree with the reviewer point that the use of a human cell line would have been better. However, the RAW 264.7 cell line was used as it was the most suitable to handle in our research facility environment. RAW 264.7 cells are easy to work with and available in large numbers, facilitating certain experiments that generally require a fairly large amount of starting material. It’s still acceptable to use RAW 264.7 cells as model in the preliminary screening of drug candidates.
Table 2: Unit has to be provided in table 2 to accurate interpret the results.
Corrected as requested
Line 193: Two fatty what? Fatty acids?
Thank you for the remark, the missing word has been added